# 3D G-CNNs for Pulmonary Nodule Detection

**Marysia Winkels**
University of Amsterdam / Aidence
marysia@aidence.com

**Taco S. Cohen**
University of Amsterdam
taco.cohen@gmail.com

## Abstract

Convolutional Neural Networks (CNNs) require a large amount of annotated data to learn from, which is often difficult to obtain in the medical domain. In this paper we show that the sample complexity of CNNs can be significantly improved by using 3D roto-translation group convolutions (G-Convs) instead of the more conventional translational convolutions. These 3D G-CNNs were applied to the problem of false positive reduction for pulmonary nodule detection, and proved to be substantially more effective in terms of performance, sensitivity to malignant nodules, and speed of convergence compared to a strong and comparable baseline architecture with regular convolutions, data augmentation and a similar number of parameters. For every dataset size tested, the G-CNN achieved a FROC score close to the CNN trained on *ten times* more data.

## 1 Introduction

Lung cancer is currently the leading cause of cancer-related death worldwide, accounting for an estimated 1.7 million deaths globally each year and 270,000 in the European Union alone [1, 2], taking more victims than breast cancer, colon cancer and prostate cancer combined [3]. This high mortality rate can be largely attributed to the fact that the majority of lung cancer is diagnosed when the cancer has already metastasised as symptoms generally do not present themselves until the cancer is at a late stage, making early detection difficult [4].

Screening of high risk groups could potentially increase early detection and thereby improve the survival rate [5, 6]. However, the (cost-) effectiveness of screening would be largely dependent on the skill, alertness and experience level of the reading radiologists, as potentially malignant lesions are easy to overlook due to the rich vascular structure of the lung (see Figure 1). A way to reduce observational oversights would be to use second readings [7, 8], a practice in which two readers independently interpret an

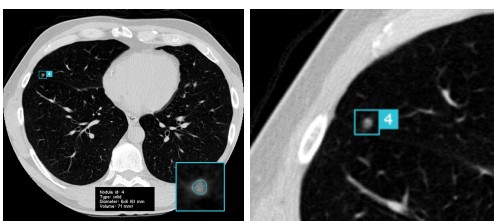

Figure 1: Lung nodule on axial thorax CT

image and combine findings, but this would also drastically add to the already increasing workload of the radiologist [9], and increase the cost of care. Thus, a potentially much more cost-effective and accurate approach would be to introduce computer aided detection (CAD) software as a second reader to assist in the detection of lung nodules [10, 11].

For medical image analysis, deep learning and in particular the Convolutional Neural Network (CNN) has become the methodology of choice. With regards to pulmonary nodule detection specifically, deep learning techniques for *candidate generation* and *false positive reduction* unambiguously outperform classical machine learning approaches [12, 13, 14]. Convolutional neural networks, however, typically require a substantial amount of labeled data to train on – something that is scarce in the medical

---

Parts of this paper appeared previously in the first author's thesis.

1st Conference on Medical Imaging with Deep Learning (MIDL 2018), Amsterdam, The Netherlands.

imaging community, both due to patient privacy concerns and the labor-intensity of obtaining high-quality annotations. The problem is further compounded by the fact that in all likelihood, many CAD systems will have to be developed for different imaging modalities, scanner types, settings, resolutions, and patient populations. All of this suggests that *data efficiency* is a major hurdle to the scalable development of CAD systems such as those which are the focus of the current work: lung nodule detection systems.

Relative to fully connected networks, CNNs are already more data efficient. This is due to the translational weight sharing in the convolutional layers. One important property of convolution layers that enables translational weight sharing, but is rarely discussed explicitly, is *translation equivariance*: a shift in the input of a layer leads to a shift in the output, $f(T\mathbf{x}) = Tf(\mathbf{x})$. Because each layer in a CNN is translation equivariant, all internal representations will shift when the network input is shifted, so that translational weight sharing is effective in each layer of a deep network.

Many kinds of patterns, including pulmonary nodules, maintain their identity not just under translation, but also under other transformations such as rotation and reflection. So it is natural to ask if CNNs can be generalized to other kinds of transformations, and indeed it was shown that by using *group convolutions*, weight sharing and equivariance can be generalized to essentially arbitrary groups of transformations [15]. Although the general theory of G-CNNs is now well established [15, 16, 17, 18], a lot of work remains in developing easy to use group convolution layers for various kinds of input data with various kinds of symmetries. This is a burgeoning field of research, with G-CNNs being developed for discrete 2D rotation and reflection symmetries [15, 19, 16], continuous planar rotations [20, 21, 22], 3D rotations of spherical signals [23], and permutations of nodes in a graph [24].

In this paper, we develop G-CNNs for three-dimensional signals such as volumetric CT images, acted on by discrete translations, rotations, and reflections. This is highly non-trivial, because the discrete roto-reflection groups in three dimensions are non-commutative and have a highly intricate structure (see Figure 3). We show that when applied to the task of false-positive reduction for pulmonary nodule detection in chest CT scans, 3D G-CNNs show remarkable data efficiency, yielding similar performance to CNNs trained on $10\times$ more data. Our implementation of 3D group convolutions is publicly available[1], so that using them is as easy as replacing `Conv3D()` by `GConv3D()`.

In what follows, we will provide a high-level overview of group convolutions as well as the various 3D roto-reflection groups considered in this work (section 2). Section 3 describes the experimental setup, including datasets, evaluation protocol network architectures, and section 4 compares G-CNNs to conventional CNNs in terms of performance and rate of convergence. We discuss these results and conclude in sections 5 and 6 respectively.

## 2 Three-dimensional G-CNNs

In this section we will explain the 3D group convolution in an elementary fashion. The goal is to convey the high level idea, focusing on the algorithm rather than the underlying mathematical theory, and using visual aids where this is helpful. For the general theory, we refer the reader to [15, 16, 17, 18].

To compute the conventional (translational) convolution of a filter with a feature map, the filter is translated across the feature map, and a dot product is computed at each position. Each cell of the output feature map is thus associated with a translation that was applied to the filter. In a group convolution, additional transformations like rotations and reflections are applied to the filters, thereby increasing the degree of weight sharing. More specifically, starting with a canonical filter with learnable parameters, one produces a number of transformed copies, which are then convolved (translationally) with the input feature maps to produce a set of output feature maps.

Thus, each learnable filter produces a number of *orientation channels*, each of which detects the same feature in a different orientation. We will refer to the set of orientation channels associated with one feature / filter as one feature map.

As shown in [15], if the transformations that are applied to the filters are chosen to form a *symmetry group $H$* (more on that later), the resulting feature maps will be equivariant to transformations from this group (as well as being equivariant to translations). More specifically, if we transform the input

---

[1]https://github.com/tscohen/GrouPy

by $h \in H$ (e.g. rotate it by 90 degrees), each orientation channel will be transformed by $h$ in the same way, *and* the orientation channels will get shuffled by a permutation matrix $\rho(h)$.

The channel-shuffling phenomenon occurs because the transformation $h$ changes the orientation of the input pattern, so that it gets picked up by a different orientation channel / transformed filter. The particular way in which the channels get shuffled by each element $h \in H$ depends on the structure of $H$ (i.e. the way transformations $g, k \in H$ compose to form a third transformation $h = gk \in H$), and is the subject of subsection 2.1.

Because of the output feature maps of a G-Conv layer have orientation channels, the filters in the second and higher layers will also need orientation channels to match those of the input. Furthermore, when applying a transformation $h \in H$ to a filter that has orientation channels, we must also shuffle the orientation channels of the filter. Doing so, the feature maps of the next layer will again have orientation channels that jointly transform equivariantly with the input of the network, so we can stack as many of these layers as we like while maintaining equivariance of the network.

In the simplest instantiation of G-CNNs, the group $H$ is the set of 2D planar rotations by $0, 90, 180$ and 270 degrees, and the whole group $G$ is just $H$ plus 2D translations. In this case, there are four orientation channels per feature, which undergo a cyclic permutation under rotation. In 3D, however, things get substantially more complicated. In the following section, we will discuss the various 3D roto-reflection groups considered in this paper, and show how one may derive the channel shuffling permutation $\rho(h)$ for each symmetry transformation $h$. Then, in section 2.2, we provide a detailed discussion of the implementation of 3D group convolutions.

## 2.1 3D roto-reflection groups

In general, the symmetry group of an object is the set of transformations that map that object back onto itself without changing it. For instance, a square can be rotated by $0, 90, 180$ and 270 degrees, and flipped, without changing it. The set of symmetries of an object has several obvious properties, such as closure (the composition $gh$ of two symmetries $g$ and $h$ is a symmetry), associativity ($h(gk) = (hg)k$ for transformations $h, g$ and $k$), identity (the identity map is always a symmetry), and inverses (the inverse of a symmetry is always a symmetry). These properties can be codified as axioms and studied abstractly, but in this paper we will only study concrete symmetry groups that are of use in 3D G-CNNs, noting only that all ideas in this paper are easily generalized to a wide variety of settings by moving to a more abstract level.

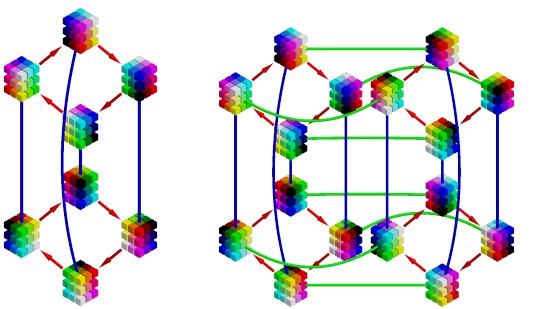

The filters in a three-dimensional CNN are not squares, but cubes or rectangular cuboids. We would restrict our study of symmetry groups to cubes, but in many 3D imaging modalities such as CT and MRI, the pixel spacing in the $x$ and $y$ directions can be different from the spacing in the $z$ direction, so that a $k \times k \times k$ filter corresponds to a spatial region that is not a cube but a cuboid with a square base.

Figure 2: Cayley diagrams of the groups $D_4$ (left) and $D_{4h}$ (right). Best viewed in color.

In addition to the cube / rectangular cuboid choice, there is the choice of whether to consider only rotations (orientation-preserving symmetries) or reflections as well. Thus, we end up with four symmetry groups of interest: the orientation-preserving and non-orientation preserving symmetries of a rectangular cuboid (called $D_4$ and $D_{4h}$, resp.), and the orientation-preserving and non-orientation-preserving symmetries of a cube ($O$ and $O_h$, resp.).

Despite the apparent simplicity of cubes and rectangular cuboids, their symmetry groups are surprisingly intricate. In figure 2 and 3, we show the *Cayley diagram* for the groups $D_4$, $D_{4h}$ and $O$ (the group $O_h$ is not shown, because it is very large). In a Cayley diagram, each node corresponds to a symmetry transformation $h \in H$, here visualized by its effect on a canonical $3 \times 3 \times 3$ filter. The diagrams also have lines and arrows of various colors that connect the nodes. Each color corresponds

to applying a particular *generator* transformation. By applying generators in sequence (i.e. following the edges of the diagram) we can make any transformation in the group. Because different sequences of generator transformations can be the equal, there can be several paths between two nodes, leading to an intricate graph structure.

For example, figure 2 (Left) shows the Cayley diagram of the group $D_4$, the group of orientation-preserving symmetries of a rectangular cuboid filter. This group is generated by the 90-degree rotation around the Z-axis (red arrow) and the 180-degree rotation around the $Y$-axis (blue line). The latter is shown as a line instead of an arrow, because it is self inverse: $h^{-1} = h$. We see that this group is not commutative, because starting from any node, following a red arrow and then a blue line leaves us in a different place from following a blue line and then a red arrow.

Similarly, figure 2 (Right) shows the Cayley diagram for $D_{4h}$, the non-orientation-preserving symmetries of a rectangular cuboid. This diagram has an additional generator, the reflection in the Z-plane (drawn as a green line). Figure 3 shows the Cayley diagram for $O$, the orientation-preserving symmetries of a cubic filter, generated by Z-axis rotations (red arrows) and rotations around a diagonal axis (blue arrows).

Recall from our previous discussion that in the implementation of the group convolution, we use the permutation $\rho(h)$ to shuffle the orientation channels of a filter in the second or higher layers. As we will now explain, we can easily derive these permutations from the Cayley diagram. We note that this only has to be done once when implementing 3D G-CNNs; when using them, this complexity is hidden by an easy to use function GConv3D().

Because in a Cayley diagram there is exactly one arrow (or line) of each color leaving from each node, these diagrams define a permutation $\rho(g)$ for each generator (edge color) $g$. This permutation maps a node to the node it is connected to by an arrow of a given color. Since every group element can be written as a composition of generators, we can obtain its permutation matrix from the permutations associated with the generators. For instance, if $g_1, g_2$ are generators

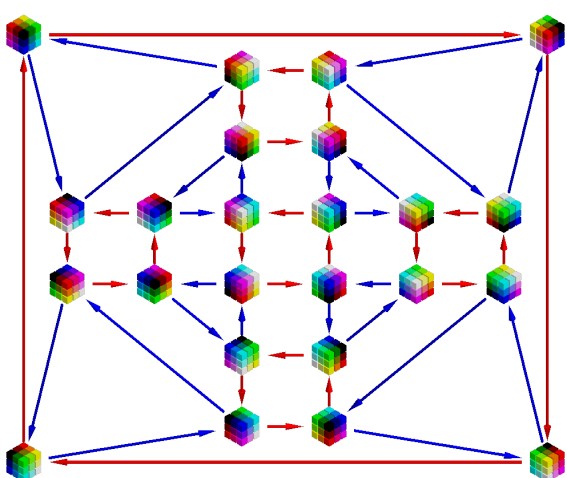

Figure 3: Cayley diagram for $O$. Red arrows correspond to Z-axis rotation, whereas blue arrows correspond to rotation around a diagonal axis. Best viewed in color.

(such as the red and blue arrows in Figure 3), then we have $\rho(g_1 g_2) = \rho(g_1)\rho(g_2)$ as the permutation associated with the node obtained by following the $g_1$ arrow after the $g_2$ arrow. Since we can read off $\rho(g_1)$ and $\rho(g_2)$ directly from the Cayley diagram, we can figure out $\rho(h)$ for all $h \in H$.

## 2.2 Implementation details

As mentioned before, the group convolution (for the first layer as well as higher layers) can be implemented in two steps: filter transformation and spatial convolution. The latter speaks for itself (simply call Conv3D(feature_maps, transformed_filters)), so we will focus on the filter transformation step.

In the first layer, the input feature maps (e.g. CT scans) and filters do not have orientation channels. Let's consider $n_0$ input channels, and $n_1$ filters (each of which has $n_0$ input channels). In the filter transformation step, we simply transform each of the $n_1$ filters by each transformation $h \in H$, leading to a bigger filter bank with $n_1 \cdot |H|$ filters (each of which still has $n_0$ input channels). When we transform a filter with $n_0$ channels by $h$, each of the channels is transformed by $h$ simultaneously, resulting in a single transformed filter with $n_0$ channels.

In the second and higher layers, we need to additionally shuffle the orientation channels of the filter by $\rho(h)$. If the input to layer $l$ has $n_l$ feature channels, each of which has $|H|$ orientation channels (for a total of $n_l \cdot |H|$ 3D channels), each of the $n_{l+1}$ filters will also have $n_l$ feature channels with

$|H|$ orientation channels each. During the filter transformation step, the filters again get transformed by each element $h \in H$, so that we end up with $n_{l+1} \cdot |H|$ transformed filters, and equally many 3D output channels.

Since the discrete rotations and reflections of a filter will spatially permute the pixels, and $\rho(h)$ permutes channels, the whole filter transformation is just the application of $|H|$ permutations to the filter bank. This can easily be implemented by an indexing operation of the filter bank with a precomputed set of indices, or by multiplying by a precomputed permutation matrix. For the details of how to precompute these, we refer to our implementation.

Considering that the cost of the filter transformation is negligible, the computational cost of a group convolution is roughly equal to the computational cost of a regular spatial convolution with a filter bank whose size is equal to the augmented filter bank used in the group convolution. In practice, we typically reduce the number of feature maps in a G-CNN, to keep the amount of computation the same, or to keep the number of parameters the same.

## 3   Experiments

Modern pulmonary nodule detection systems consist of the following five subsystems: data acquisition (obtaining the medical images), preprocessing (to improve image quality), segmentation (to separate lung tissue from other organs and tissues on the chest CT), localisation (detecting suspect lesions and potential nodule candidates) and false positive reduction (classification of found nodule candidates as nodule or non-nodule). The experiments in this work will focus on false positive reduction only. This reduces the problem to a relatively straightforward classification problem, and thus enables a clean comparison between CNNs and G-CNNs, evaluated under identical circumstances.

To determine whether a G-CNN is indeed beneficial for false positive reduction, the performance of networks with G-Convs for various 3D groups G (see subsection 2.1) are compared to a baseline network with regular 3D convolutions. To further investigate the data-efficiency of CNNs and G-CNNs, we conduct this experiment for various training dataset sizes varying from 30 to 30,000 data samples. In addition, we evaluated the convergence speed of G-CNNs and regular CNNs, and found that the former converge substantially faster.

### 3.1   Datasets

The scans used for the experiments originate from the NLST [5] and LIDC/IDRI [25] datasets. The NLST dataset contains scans from the CT arm of the National Lung Screening Trial, a randomized controlled trial to determine whether screening with low-dose CT (without contrast) reduces the mortality from lung cancer in high-risk individuals relative to screening with chest radiography. The LIDC/IDRI dataset is relatively varied, as scans were acquired with a wide range of scanner models and acquisition parameters and contains both low-dose and full-dose CTs taken with or without contrast. The images in the LIDC/IDRI database were combined with nodule annotations as well as subjective assessments of various nodule characteristics (such as suspected malignancy) provided by four expert thoracic radiologists. Unlike the NLST, the LIDC/IDRI database does not represent a screening population, as the inclusion criteria allowed any type of participant.

All scans from the NLST and LIDC/IDRI datasets with an original slice thickness equal to or less than 2.5mm were processed by the same candidate generation model to provide center coordinates of potential nodules. These center coordinates were used to extract $12 \times 72 \times 72$ patches from the original scans, where each voxel represents $1.25 \times .5 \times .5$mm of lung tissue. Values of interest for nodule detection lie approximately between -1000 Hounsfield Units (air) and 300 Hounsfield Units (soft-tissue) and this range was normalized to a $[-1, 1]$ range.

Due to the higher annotation quality and higher variety of acquisition types of the LIDC/IDRI, along with the higher volume of available NLST image data, the training and validation is done on potential candidates from the NLST dataset and testing is done on the LIDC/IDRI nodule candidates. This division of datasets, along with the exclusion of scans with a slice thickness greater than 2.5mm, allowed us to use the reference standard for nodule detection as used by the LUNA16 grand challenge [26] and performance metric as specified by the ANODE09 study [27]. This setup results in a total of $30,000$ data samples for training, $8,889$ for validation, and $8,582$ for testing. Models are trained on subsets of this dataset of various sizes: 30, 300, 3,000 and 30,000 samples. Each training set is

| Set | Source | Candidates | Positive % | Negative % |
|---|---|---|---|---|
| *Training* | NLST | *max.* 30,000 | 50.0 | 50.0 |
| *Validation* | NLST | 8,889 | 20.6 | 79.4 |
| *Test* | LIDC/IDRI | 8,582 | 13.3 | 86.7 |

Table 1: Specifics of the training, validation and test set sizes and class ratios.

balanced, and each smaller training set is a subset of all larger training sets. The details of the train, validation and test sets are specified in Table 1.

## 3.2 Network architectures & training procedure

A baseline network was established with 6 convolutional layers consisting of $3 \times 3 \times 3$ convolutions, batch normalization and ReLU nonlinearities. In addition, the network uses 3D max pooling with same padding after the first, third and fifth layer, dropout after the second and fourth layer, and has a fully-connected layer as a last layer. We refer to this baseline network as the $\mathbb{Z}^3$-CNN, because, like every conventional 3D CNN, it is a G-CNN for the group of 3D translations, $\mathbb{Z}^3$. The $\mathbb{Z}^3$-CNN baseline, when trained on the whole dataset, was found to achieve competitive performance based on the LUNA16 grand challenge leader board, and therefore deemed sufficiently representative of a modern pulmonary nodule CAD system.

The G-Conv variants of the baseline were created by simply replacing the 3D convolution in the baseline with a G-Conv for the group $D_4, D_{4h}, O$ or $O_h$ (see section 2.1). This leads to an increase in the number of 3D channels, and hence the number of parameters per filter. Hence, the number of desired output channels ($n_{l+1}$) is divided by $\sqrt{|H|}$ to keep the number of parameters roughly the same and the network comparable to the baseline.

We minimize the cross-entropy loss using the Adam optimizer [28]. The weights were initialized using the uniform Xavier method [29]. For training, we use a mini-batch size of 30 (the size of the smallest training set) for all training set sizes. We use validation-based early stopping. A single data augmentation scheme (continuous rotation by $0 - 360^o$, reflection over all axes, small translations over all axes, scaling between $.8 - 1.2$, added noise, value remapping) was used for all training runs and all architectures.

## 3.3 Evaluation

Despite the availability of a clear definition of a lung nodule (given by the Fleischer Glossary), several studies confirm that observers often disagree on what constitutes a lung nodule [30, 31, 32]. This poses a problem in the benchmarking of CAD systems.

In order to deal with inter-observer disagreements, only those nodules accepted by 3 out of four radiologists (and $\geq$ 3mm and $\leq$ 30mm in largest axial diameter) are considered essential for the system to detect. Nodules accepted by fewer than three radiologists, those smaller than 3mm or larger than 30mm in diameter, or with benign characteristics such as calcification, are ignored in evaluation and do not count towards the positives or the negatives. The idea to differentiate between relevant (essential to detect) and irrelevant (optional to detect) findings was first proposed in the ANODE09 study [27].

ANODE09 also introduced the Free-Response Operating Characteristic (FROC) analysis, where the sensitivity is plotted against the average number of false positives per scan. FROC analysis, as opposed to any single scalar performance metric, makes it possible to deal with differences in preference regarding the trade-off between sensitivity and false positive rate for various users. We use this to evaluate our systems. To also facilitate direct quantitative comparisons between systems, we compute an overall system score based on the FROC analysis, which is the average of the sensitivity at seven predefined false positive rates ($\frac{1}{8}; \frac{1}{4}; \frac{1}{2}; 1; 2; 4;$ and 8).

This evaluation protocol described in this section is identical to the method used to score the participants of the LUNA16 nodule detection grand challenge [26], and is the de facto standard for evaluation of lung nodule detection systems.

# 4 Results

## 4.1 FROC analysis

Figure 4 shows the FROC curve for each $G$-CNN and training set size. Table 2 contains the overall system score for each group and training set size combination, and has the highest (**bold**) and lowest (*italic*) scoring model per training set size highlighted.

| N | $\mathbb{Z}^3$ | $D_4$ | $D_{4h}$ | $O$ | $O_h$ |
|---|---|---|---|---|---|
| 30 | *0.252* | 0.398 | 0.382 | **0.562** | 0.514 |
| 300 | *0.550* | 0.765 | 0.759 | **0.767** | 0.733 |
| 3,000 | *0.791* | 0.849 | 0.844 | 0.830 | **0.850** |
| 30,000 | *0.843* | 0.867 | **0.880** | 0.873 | 0.869 |

Table 2: Overall score for all training set sizes $N$ and transformation groups $G$. The group $G = \mathbb{Z}^3$ corresponds to the standard translational CNN baseline.

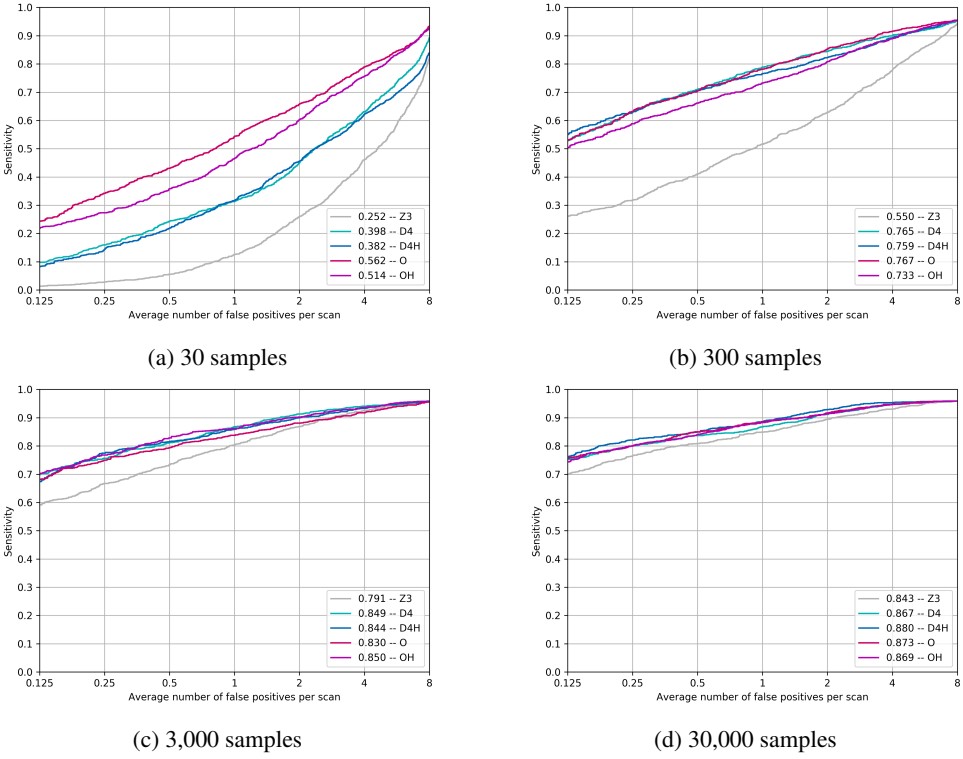

Figure 4: FROC curves for all groups per training set size.

## 4.2 Rate of convergence

Figure 5 plots the training loss per epoch, for training runs with dataset size 3,000 and 30,000. Table 3 lists the number of training epochs required by each network to achieve a validation loss that was at least as good as the best validation loss achieved by the baseline.

| $N$ | $\mathbb{Z}^3$ | $D_4$ | $D_{4h}$ | $O$ | $O_h$ | total epochs |
|---|---|---|---|---|---|---|
| 3,000 | *82* | 33 | 22 | 21 | **11** | *100* |
| 30,000 | *41* | 4 | 9 | 7 | **3** | *50* |

Table 3: Number of epochs after which the loss is equal to or lower than the lowest validation loss achieved on the baseline for each group.

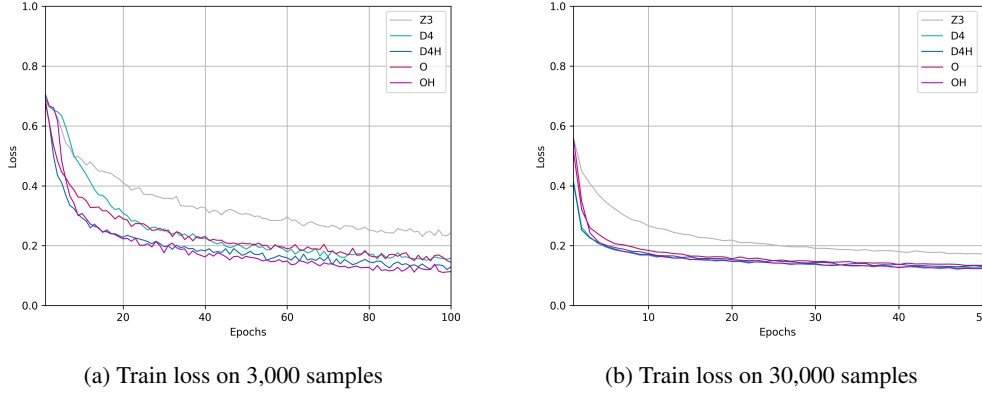

(a) Train loss on 3,000 samples          (b) Train loss on 30,000 samples

Figure 5: Learning curves for all networks trained on 3,000 and 30,000 samples.

## 5    Discussion & future work

The aggregated system scores in Table 2 and FROC analysis per training set size in Figure 4 show that not only do *all* G-CNNs outperform the baseline when trained on the same training set, they also regularly outperform the baseline trained on $10\times$ the data. To take but one example, the $O$-CNN trained on $N$ samples performs similarly to the baseline $\mathbb{Z}^3$-CNN trained on $10\cdot N$ samples. Hence we can say that $G$-CNNs were found to be approximately $10\times$ more data efficient, with the differences being more pronounced in the small data regime.

Variation in scores for the G-CNNs on most dataset sizes are largely negligible and can be attributed to different weight initialisations, random data augmentation, or other small variations within the training process. Remarkably though, there was a notable difference in performance between the groups of octahedral symmetry ($O$, $O_h$) and rectangular cuboid symmetry ($D_4$, $D_{4h}$) for the really small dataset size of 30 samples. Although in most cases there does not appear to be a group that consistently outperforms the other, this finding does indicate that more empirical research into effect of the group (order and type) on performance may be necessary.

Moreover, G-CNNs seem to require fewer epochs to converge and generalise than regular CNNs. From Figure 5, we can see that the group-convolutional models show a faster decline in training loss within the early stages of training compared to the baseline. This faster convergence can be explained by the fact that each parameter receives a gradient signal from multiple 3D feature maps at once. Additionally, as shown in Table 3, G-CNNs typically take only a few epochs to reach a validation loss that is better than the best validation loss achieved by the baseline. For example, the baseline $\mathbb{Z}^3$-CNN took 41 epochs to achieve its optimal validation loss (when trained on 30,000 samples), whereas the $O_h$-CNN reached the same performance in just 3 epochs. It should be noted however that the advantage of faster convergence and generalisation is partly negated by the fact that the processing of each epoch typically takes longer for a G-CNN than for a conventional CNN, because the G-CNN has more 3D channels given a fixed parameter budget.

For the time being, the bottleneck of group-convolutional neural networks will most likely be the GPU memory requirements, as limits to the GPU resources may prevent the optimal model size from being reached. Fortunately, this issue can be resolved by further optimisations to the code, multi-GPU training, and future hardware advances.

## 6    Conclusion

In this work we have presented 3D Group-equivariant Convolutional Neural Networks (G-CNNs), and applied them to the problem of false positive reduction for lung nodule detection. 3D G-CNN architectures – obtained by simply replacing convolutions by group convolutions – unambiguously outperformed the baseline CNN on this task, especially on small datasets, without any further tuning. In our experiments, G-CNNs proved to be about $10\times$ more data efficient than conventional CNNs. This improvement in statistical efficiency corresponds to a major reduction in cost of data collection, and brings pulmonary nodule detection and other CAD systems closer to reality.

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

# A  Sensitivity to malignant nodules

The primary motivation for nodule detection is to find potentially malignant nodules. This appendix highlights the results of the systems performances with respect to malignancy. A nodule is considered *malignant* for this exercise if at least three out of four radiologists suspected the nodule to be moderately or highly suspicious and no radiologists indicated the nodule to be moderately or highly unlikely to be malignant. In total, 129 nodules qualify as malignant according to this constraint.

To evaluate the sensitivity of the system towards specifically malignant nodules, we consider a set number of true positives, and evaluate what number of these true positives is not only a nodule, but also malignant. Table 4 provides an overview of the number of malignant nodules in the $n$ true positives that received the highest estimated probability for each model, where $n \in \{100, 150, 250\}$.

For example, out of the 100 true positive nodules that were deemed most likely to be nodules by the baseline model trained on 30,000 data samples, 13 were malignant.

| $N$ | $n$ | $\mathbb{Z}^3$ | $D_4$ | $D_{4h}$ | $\mathbf{O}$ | $O_h$ |
|---|---|---|---|---|---|---|
| | 100 | *5* | 21 | 25 | 20 | **37** |
| 3,000 | 150 | *6* | 37 | 40 | 25 | **45** |
| | 250 | *14* | 59 | 59 | 51 | **62** |
| | 100 | *13* | 31 | **45** | 44 | 32 |
| 30,000 | 150 | *18* | 49 | 60 | **61** | 38 |
| | 250 | *31* | 61 | **77** | **77** | 63 |

Table 4: Number of malignant nodules from the $n$ true positives that received the highest probability estimation.

The group-convolution based models seem considerably more sensitive to malignant nodules in their top rated true positives. The number of malignant nodules in the 250 top probability true positives as judged by the baseline system, are equal to or less than the number of malignant nodules in the top 100 for the group-convolutional systems. More specifically, whereas there were 14 (out of 129) malignant nodules in the top 250 nodules for $\mathbb{Z}^3$ trained on 3,000 data samples, there were more malignant nodules in the top 100 for any of the g-convolutional models trained on the same dataset.

