# OpenReview forum: "3D G-CNNs for Pulmonary Nodule Detection"
_MIDL.amsterdam/2018/Conference — MIDL 2018 Oral_

### Review · AnonReviewer2 · 2018-05-08
**intuitive idea, good results**

**Rating:** 4
**Confidence:** 2

**Review:**

The study evaluates CNNs with a different type of convolutional filters. The idea is intuitive and results illustrate an impressive speed up in training time. Clinical application is nodule detection.

Pro: This is an interesting study, featuring a CNN modification that will be of interest across application domains and for many attendees of the conference.

Con: Lung nodules have symmetric properties that may be a particularly good match for the type of network modification described here. If accepted, maybe the authors will comment?

**Special Issue:**

Yes

---

> ### Comment · ~Taco_Cohen1 · 2018-05-14
> **Thanks**
>
> Thank you for your review. Regarding the applicability to other datasets, please see our reply to AnonReviewer1. We will add a discussion on the applicability to other datasets to the paper.

---

### Review · AnonReviewer3 · 2018-05-08
**The authors propose an interesting extension of 3D CNN by replacing general convolutional kernels with groups of convolutions, in which multiple transformations (rotation & flip) of each general kernel is considered to enhance the degree of weight sharing. The proposed group convolutional neuro networks (G-CNNs) are then applied to detecting pulmonary nodule in CT scans, showing competitive performance compared with general CNN, especially when the training data is limited.**

**Rating:** 3
**Confidence:** 3

**Review:**

While the motivation of including additional transformations (e.g., rotations and reflections) of convolutional kernels is easy to follow, it is not clear if the limited transformations considered in this paper could largely cover real situations.

It is interesting to see if the general CNN could have similar performance compared with G-CNN by including simple data augmentation (e.g., random rotation & flipping) operations during training, especially in the case of small training data (e.g., 30 & 300).

Based on Table 2, we could observe that D_{4h} performed even worse than D_4 in the case of limited training samples (e.g., 30 & 300). This partially contradicts with the authors’ assumption, since D_{4h} should be more powerful than D_4 in dealing with small training data due to more complicated transformations were considered.

The authors present four variants of their G-CNN, while it is hard to determine which one should be considered or selected from the practical application view.


**Special Issue:**

No

---

> ### Comment · ~Taco_Cohen1 · 2018-05-14
> **Data augmentation & group size**
>
> We would like to thank the reviewer for their detailed review and comments. We agree with the reviewer that it is a priori not clear if including a limited set of transformations, such as discrete rotations and reflections, would help generalization, since they cover only part of the transformations that one might expect in real data. Nevertheless, we believe that our experimental results strongly suggest that this is indeed very effective; we have shown that the 3D G-CNN is no less than ten times more data efficient than a conventional CNN.
>
> Importantly, we note that as mentioned in the abstract and section 3.2, we used data augmentation in all experiments, for all models (both CNNs and G-CNNs). Specifically, we used continuous rotation by 0 − 360 degrees, reflection over all axes, small translations over all axes, scaling between .8 − 1.2, added noise, and value remapping. This shows that the improvements in the results observed for the G-CNN really cannot be achieved using conventional data augmentation.
>
> From these results, we can conclude that even with data augmentation, it is hard for a conventional CNN to learn exact equivariance. This is perhaps not so surprising, given that a fully connected network could also learn translation equivariance if trained with translational augmentation, but still we prefer to use convolutional networks instead, which hard-wire the translational equivariance & weight sharing.
>
> Another thing that appears to benefit the G-CNN is it's improved optimizability (section 4.2 & table 3), which stems from its increased weight sharing.
>
> It is true that the results do not unambiguously support the hypothesis that bigger symmetry groups are always better. However, the difference between D4 and D4h for the N=30 case (0.398 vs 0.382) seems hardly significant, and can easily be attributed to random fluctuations. The difference to the larger groups O and Oh (0.562 and 0.514) is much more substantial.
>
> Nevertheless, we do not believe or claim that bigger symmetry groups are always better. The reason is that as the size of the group is increased, the number of parameters per filter also increases. Hence, in order to keep the number of parameters the same, the number of distinct filters is reduced. A network with a very large symmetry group can therefore detect fewer truly different patterns (although it can recognize them in more orientations). We believe that this tradeoff explains the results. We thank the reviewer for raising this issue, and will include a discussion of this issue in the paper.

---

### Review · AnonReviewer1 · 2018-05-09
**On this paper the authors extend G-CNNs to the group of transformations consisting in rotations and reflections in 3D, and apply this to lung nodule detection on 3D CT scans.**

**Rating:** 4
**Confidence:** 2

**Review:**

The paper is in general well written, and even though section 2.1 can be considered a challenge to be explained, its figures make the reader build intuition about it. The experimental section looks very convincing.
The choice of the dataset fits quite well the assumptions made by the paper, but in general, it would be interesting to have some discussion as to when this kind of approach could be applied. Submitting this paper to this conference is also germane given the amount of 3d scans in the field (with the caveat that invariance to all 3d transformations may not apply), so I recommend to accept this paper.

**Special Issue:**

Definitely

---

> ### Comment · ~Taco_Cohen1 · 2018-05-14
> **Applicability**
>
> Thank you for your review and positive comments. Lung nodule CT scans are indeed ideally suited for our method, but we believe that the method can also work on data with approximate symmetry. For instance, in the original paper on 2D G-CNNs [1], it was found that they work well for CIFAR, which is not rotation invariant (objects always appear upright). That the method still works in this case is likely due to the fact that small-scale details still *do* appear in every orientation, so rotational weight sharing helps, at least in the first layers.
>
> [1] T.S. Cohen & Max Welling, Group Equivariant Convolutional Networks, ICML 2016.

---

### Comment · ~Bram_van_Ginneken1 · 2018-05-18
**Selection for longlist for special issue Medical Image Analysis**

Dear authors,

Congratulations on your acceptance to MIDL! We have selected your paper on the longlist for the Medical Image Analysis Special Issue. Please read this page:
https://midl.amsterdam/special-issue-in-medical-image-analysis/
Please answer the three questions that are listed on that page about your interest in submitting to the special issue, potential overlap with other publications, and related publications.

You can post your answer here directly below on openreview.net, or mail me directly at bram.vanginneken@radboudumc.nl.

Best regards, Bram

---

### Decision · Program_Chairs · 2018-05-15
**Paper26 Acceptance Decision**

Oral